# Comment on Mitteregger et al. A Variant Carbapenem Inactivation Method (CIM) for *Acinetobacter baumannii* Group with Shortened Time-to-Result: rCIM-A. *Pathogens* 2022, *11*, 482

**DOI:** 10.3390/pathogens11070751

**Published:** 2022-07-01

**Authors:** Andrei-Alexandru Muntean, Madalina-Maria Muntean, Saoussen Oueslati, Remy Bonnin, Thierry Naas, Mircea Ioan Popa

**Affiliations:** 1Cantacuzino National Medical Military Institute for Research and Developement, Carol Davila University of Medicine and Pharmacy, 050096 Bucharest, Romania; alexandru.muntean@umfcd.ro (A.-A.M.); mmada.muntean@gmail.com (M.-M.M.); 2University of Paris-Saclay, Kremlin-Bicetre Hospital, 94270 Le Kremlin-Bicêtre, France; oueslati.saoussen@gmail.com; 3University of Paris-Saclay, Kremlin Bicetre Hospital, Associated French National Reference Center for Antibiotic Resistance: Carbapenemase-Producing Enterobacteriaceae, 94270 Le Kremlin-Bicêtre, France; remy.bonnin@u-psud.fr (R.B.); thierry.naas@aphp.fr (T.N.)

We have read the article published by Mitteregger et al. [1], which describes a “novel” variant of the Carbapenem Inactivation Method (CIM) protocol for *Acinetobacter* spp. and feel that the chosen name for this protocol, the rCIM-A, leads to confusion (as this name is already used) and that the work lacks novel aspects and may be difficult to interpret.

The publication by Mitteregger et al. [1] evaluates a proposed protocol for detection of carbapenemase-producing *Acinetobacter* spp. in a same-day fashion (as the authors state, a “shortened time-to-result”). On this basis, the authors have chosen to name their proposed protocol “the rapid CIM for *Acinetobacter* spp. (rCIM-A)”. While there are numerous variants of CIM that have been proposed [1,2], the “rCIM-A” name has already been published almost a year ago to define a rapid CIM (rCIM) protocol that optimizes results for chromosomal cephalosporinase (AmpC) hyper-producing Enterobacterales [3]. Moreover, we consider that the protocol proposed is not truly “rapid”. The protocol by Mitteregger et al. [1] is similar to the original Carbapenem Inactivation Method published in 2015 by Kim van der Zwaluw et al. [4]. In this latter paper, the authors proposed a 2-h incubation of the Meropenem disc with the bacteria to be tested in water. Subsequently, the Meropenem disk is retrieved, and placed on an *E. coli* ATCC 25922 inoculated Mueller Hinton plate. Inhibition of indicator *E. coli* could be read after six hours, or overnight, according to the laboratory’s setup and needs. In 2017, Gauthier et al. [5] published an evaluation by the French National Reference Center of the CIM, noting that results were mostly readable after overnight incubation, thus suggesting an optional reading being carried out at 6 h with a mandatory final reading carried out after overnight incubation to confirm the early reading results. We find little evidence of protocol optimization which would lead to better growth of the indicator strain, as this step of the test is virtually identical.

The Carbapenem Inactivation Method became “rapid” with the work by Muntean et al. [6] with final test results in less than 3 h, a time-to-result similar to those of other rapid CPE confirmation methods (colorimetric, spectrometry, etc.), thus changing the paradigm of CIM testing. The protocol published by Muntean et al. differed substantially from the original protocol published by Kim van der Zwaluw et al. [4]. Firstly, the incubation time of the bacteria to be tested with the antibiotic was shortened to 30 min (as opposed to 2 h). This protocol also implies the use of the supernatant, in which the antibiotic diffused, with no further need to retrieve the antibiotic disk. In the second step, the *E. coli* ATCC 25922 growth indicator cultured is cultured in a liquid medium (as opposed to streaking on a Mueller Hinton plate). The indicator is challenged with the previously mentioned supernatant, and the results are evaluated through nephelometric reading after 1.5 and 2 h growth. In 2021, we expanded on the rCIM protocol, the rCIM-A that was optimized to be able to distinguish true CPEs from AmpC hyper-producers (that sometimes give false positive results), by adding Cloxacillin, which inhibits AmpCs. Thus, the rCIM-A showed improved specificity, while remaining “rapid” (less than 3 h) [3].

Thus, we consider that the protocol published by Mitteregger et al. [1], of a shortened time-to-result CIM protocol for *Acinetobacter* spp. is in concordance to the initial paper of Kim van der Zwaluw et al. [4] and argue that the rCIM nomenclature should be used for carbapenem inactivation tests where the supernatant is used to challenge the indicator strain grown in a liquid medium and the results are obtained in just a couple of hours, with no overnight growth.

Another possible important point pertains to the description of what the authors call “carryover microsatellites”, which the authors state are a “new category”, a “currently not yet reported phenomenon”, which could lead to modification of the classification criteria-as their presence could identify true carbapenemase producers even when the cut-off was greater than that calculated [1]. The authors fail to mention if any strains were reclassified in accordance with this observation. This is not a new phenomenon, as it was reported by Pierce et al. in 2017, in the initial description of mCIM [7]. Therein, the presence of microcolonies within the proposed “indeterminate” zone (16–18 mm) classified the investigated strain as being positive; similarly, presence of microcolonies within the inhibition zones of 19 mm or more classified the strain as indeterminate. When testing non-fermenter strains, the evaluators did not identify the same importance of the phenomenon, but this could be due to available tested strains [8]. While the authors of the mCIM did not state the nature of these microcolonies, it seems that the phenomenon is clearly presented.

Mitteregger et al. [1] make the claim that “the assay performance would not be strongly negatively affected if the three (carbapenem) disks were incubated in one and the same strain suspension”. This seems speculative at best, as beta-lactamases are, to some extent, an expendable commodity, and differing expression rates and affinities may influence the activity of weak carbapenemases in the presence of even a single carbapenem [9]. This has been clinically exploited in trials of dual beta-lactam therapy [10].

Lastly, we would remark that Mitteregger et al. [1] fail to state a clear protocol for step-by-step interpretation of the test. For the 6 h-time point, they suggest the interpretation of the Ertapenem disk, according to the presence or absence of an inhibition zone around the disk. The absence of an inhibition zone was associated with 100% specificity, meaning the presence of a true carbapenemase producer. The presence of a zone of inhibition around the Ertapenem disk seems to require reading of the Imipenem disk which offers best sensitivity. Then, a reference is made to checking the Ertapenem and Meropenem disks at 16–18 h, to confirm specificity (offering a cut-off for Ertapenem and Meropenem of 26 and 25.5 mm, respectively). This seems particularly difficult and unnecessary, for a number of reasons: (1) Would the “carryover microsatellites”/“microcolonies” be distinctly visible after only 6 h incubation and does that negatively impact reporting (specifically, reclassification)? Is this phenomenon something that was also recorded around the Imipenem disk, and has this impacted classification? A table showing results from which the reader could possibly derive the authors’ experience is surely lacking. As stated by Gauthier et al. interpretation of growth of *E. coli* ATCC 25922 on Mueller-Hinton agar in relation to the antibiotic is not always interpretable at 6 h [5]. The supplementary images seem to suggest one to two “carryover microsatellites” colonies as being diagnostic, which we feel requires a very high level of confidence from the interpreter. For an unknown reason, the authors chose, for the ultimately proposed incubation medium (TSB-Triton-X-100–0.1%), an image in which only 30 min of incubation seems to have been carried out. Perhaps, in this case, the plate with the Imipenem disk would be diagnostic, but the data and the images are not presented. (2) There is a lack of standardized internal controls (positive control, negative control, antibiotic disk control), of an expected range of zones of inhibition after incubation of the carbapenem disks, and a lack of consideration for medium and antibiotic differences [11,12]. This makes an indication of a cut-off of 25.5 mm look like overfitting of data, and not something that can be useful for the practitioner. One only needs to look at the results of a multicenter study for the (arguably simpler) interpretation of mCIM to see the discrepancies in interpretation and the need for very clear interpretation criteria and training [7,8].

Therefore, we suggest that the name of the published test by Mitteregger et al. [1], should be changed to CIM-Ab, given the fact that the rCIM-A acronym is already published and that the time-to result is comparable to that published in the original paper by Kim van der Zwaluw et al. [4]. Using the same name, for two very different CIM based assays will shed confusion in the field of diagnostics. We would also suggest that the authors reconsider the presentation of the “carryover microsatellites” phenomenon (“microcolonies” as per Pierce et al.) as novel, citing missing papers and offering a step-by-step interpretation guide for medical microbiologists.

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
