# Peer review of "Comment on Mitteregger et al. A Variant Carbapenem Inactivation Method (CIM) for Acinetobacter baumannii Group with Shortened Time-to-Result: rCIM-A. Pathogens 2022, 11, 482"

_pathogens, 2022, doi:10.3390/pathogens11070751_

Round 1

Reviewer 1 Report

The manuscript by Alexandru Andrei Muntean et al. highlights very interesting comments on inaccuracies in the manuscript entitled "A Variant Carbapenem Inactivation Method (CIM) for Acinetobacter baumannii Group with Shortened Time-to-Result: rCIM-A".

I recommend publication after minor modification ("et al." in italics).

Author Response

Dear Reviewer,

We thank the reviewer for the kind appraisal.

The suggested have been performed on the manuscript.

Kind regards,

The authors

Reviewer 2 Report

In this comment, Muntean et al are raising several technical problems and an inappropriate method name developed by Mitteregger et al,. Based on Muntean et al, the same name has been given by them for a different purpose.

While the manuscript needs an extensive English edition, this comment seems legit and adds explanations to Mitteregger et al, and therefore needs edition.

Line 19: “as far as the article describes”, not needed.

Lines 23-24: rephrase

Lines 24-25: not clear

Line 26: rephrase

Line 31: rephrase

Line 33: rephrase

Line 34: correct the typo (..)

Line 38-39: not clear

Line 41: remove “.” After ref 6.

Lines 41-42: not clear. Muntean et al (ref 6) take 3 hours or 30 minutes?

Line 53: meaning of “initial reading”?

Line 72: “(apparently unverified)”, not needed or need to be tempered.

Line 114: “cite the relevant literature”, I would replace by “need to cite missing papers” as many references were cited with relevant discussion in Mitteregger et al,. Globally, I recommend avoiding unnecessary negative words like cumbersome. 

Author Response

Dear Reviewer,

Thank you for the time to review our work.

We will address your comments and suggestions one by one in what follows:

  • While the manuscript needs an extensive English edition, this comment seems legit and adds explanations to Mitteregger et al, and therefore needs edition.

We thank you the Reviewer for their appreciation. English was reviewed, the manuscript was carefully read by the authors and an English native in order to address English language usage and readability.

  • Line 19: “as far as the article describes”, not needed

We thank the Reviewer for drawing our attention to this. We have edited in order to eliminate this phrase.

  • Lines 23-24: rephrase
  • Lines 24-25: not clear
  • Line 26: rephrase

The section which contained Lines 23-26 was adapted to eliminate unnecessary phrases, to indicate the literature review done by Mitteregger et al., to eliminate repetition and make more readable.

  • Line 31, 33: rephrase, Line 34: correct the typo (..)

We thank the Reviewer. This is Edited to eliminate

  • Line 38-39: not clear

This section was rewritten to better convey the ideas.

  • Line 41: remove “.” After ref 6.

We thank the Reviewer for their diligence. The extra fullstop was removed.

  • Lines 41-42: not clear. Muntean et al (ref 6) take 3 hours or 30 minutes?

We thank the Reviewer for raising the issue. This point was extensively reviewed to make clear. All CIM tests are two-step, with a first step in order to (potentially) inactivate the carbapenem and another step to evaluate the activity. rCIM (Muntean et al, ref 6) consists of a 30 minute first incubation step and 2 hours second step. Total hands-on time, under 3 hours. 

  • Line 53: meaning of “initial reading”?

We thank the Reivewer for pointing this out. This was corrected to „initial paper” and the first author's name was provided in full

  • Line 72: “(apparently unverified)”, not needed or need to be tempered.

We thank the Reviewer for their correct appreciation of this statement. It was eliminated.

  • Line 114: “cite the relevant literature”, I would replace by “need to cite missing papers” as many references were cited with relevant discussion in Mitteregger et al.

We thank the Reviewer and welcome their input. We have adapted the text.

  • Globally, I recommend avoiding unnecessary negative words like cumbersome. 

We thank the Reviewer for his valuable advice and moderation.

We have so adapted the manuscript to take into account this recommendation.